# Circle(s) of Life: The Circadian Clock from Birth to Death

**DOI:** 10.3390/biology12030383

**Published:** 2023-02-28

**Authors:** Iwona Olejniczak, Violetta Pilorz, Henrik Oster

**Affiliations:** 1Institute of Neurobiology, University of Lübeck, 23562 Lübeck, Germany; 2Center of Brain, Behavior, and Metabolism, University of Lübeck, 23562 Lübeck, Germany

**Keywords:** circadian clock, prenatal development, infancy, childhood, adolescence, adulthood, aging

## Abstract

**Simple Summary:**

The circadian clock is a prominent regulator of physiology. Most studies so far have investigated its role in health and disease in adult subjects and animals. However, circadian clock characteristics change over one’s lifetime. This review describes how the clock develops during pregnancy and infancy, how it evolves in childhood and adolescence and, finally, how its function is impacted by aging, with a special focus on the female reproductive system. Understanding those changes can contribute to adapting behavioral and medical interventions to the patients’ age and sex-specific needs.

**Abstract:**

Most lifeforms on earth use endogenous, so-called circadian clocks to adapt to 24-h cycles in environmental demands driven by the planet’s rotation around its axis. Interactions with the environment change over the course of a lifetime, and so does regulation of the circadian clock system. In this review, we summarize how circadian clocks develop in humans and experimental rodents during embryonic development, how they mature after birth and what changes occur during puberty, adolescence and with increasing age. Special emphasis is laid on the circadian regulation of reproductive systems as major organizers of life segments and life span. We discuss differences in sexes and outline potential areas for future research. Finally, potential options for medical applications of lifespan chronobiology are discussed.

## 1. Introduction

An organism needs to interact with its surroundings in search of nutrition, to protect itself from predation, and to seek mates. In addition to reacting to environmental changes, anticipation of such conveys survival benefits. Most lifeforms on earth are subjected to a 24-h day-night cycle driven by the planet’s rotation around its axis. An intrinsic biological timing tool, called the circadian clock, keeps track of this rhythm. Mechanistically, the circadian clock is generated by a transcriptional–translational feedback loop. The exact players vary between species, but in mammals the core clock loop consists of the brain and muscle ARNT-like 1—circadian locomotor output cycles kaput (BMAL1-CLOCK) dimer as positive regulator and Period (PER1-3) and Cryptochrome (CRY1/2) complexes as negative regulators. Additional proteins such as Reverse erythro-blastoma (REV-ERBα/β), RAR-related orphan receptor (RORα-γ), D-box binding PAR bZIP transcription factor (DBP) and Nuclear factor, interleukin 3 regulated (NFIL3) form stabilizing loops [1]. Together with inputs through post-translational modifications of these clock proteins involving Casein kinase 1 and Sirtuin 1 (CK1δ/ε and SIRT1, respectively, refs. [2,3]) they create a ~24-h oscillation. The main circadian pacemaker is located in the suprachiasmatic nucleus (SCN) of the hypothalamus, but functional clocks are expressed in almost all tissues and cells [4,5]. While principally autonomous, this clock network receives signals from the environment that realign its oscillations with the external light-dark cycle on a daily basis in a process called entrainment. Light is the most potent entrainment signal (or zeitgeber, reviewed in [6,7]), but others—e.g., food intake, exercise—also synchronize the clock [8,9,10].

Most circadian research on mammals has been conducted on adults—usually between 3–6 months of age in rodents or 20–40 years in human subjects. However, it is known that the properties of the circadian clock vary with age. Both phase and strength (amplitude) of the clock change over one’s lifetime. As circadian dysfunction is implicated in the pathophysiology of numerous diseases, including obesity, cancer, mood and neurodegenerative disorders [11,12,13], understanding the function of the clock at different stages of life can contribute to improving treatments and developing novel preventive strategies. In this review, we follow the circadian clock through the circle of life.

## 2. Prenatal and Infancy

Prenatal period—A first description of circadian clocks during development was provided by Reppert and Schwartz in 1983 [14], who elucidated daily changes in glucose utilization in mouse embryos three days before gestation. These glucose rhythms reflected the maternal phase. Subsequent studies on mothers with Bmal1 deficiency and SCN lesions confirmed this finding [14,15,16]. Intriguingly, studies on female mice with double clock gene mutations showed that the fetal clock rhythm remains unchanged [17]. This indicates that the pups are capable of developing autonomous circadian rhythms even without central and maternal clocks and the maternal clock helps to synchronize the fetal clock during gestation. Hence, the development of the embryonal clock relies strongly on the maternal circadian system to set the environmental context and relay time of day information [18,19]. From the start of fertilization, the embryo relies on maternal nutrition and communication via several signals, including hormones, to be best prepared for life outside of the utero. To meet the required conditions for the developing fetus, changes in maternal physiology during pregnancy are necessary. These changes involve release rhythms and concentrations of hormones, e.g., glucocorticoids (GCs) and melatonin, which are conveyed via and, in part, controlled by the placenta (Figure 1) (the interface between fetal and maternal circulatory systems) [18,20,21].

Melatonin—To date, melatonin and GCs are the best studied hormones in relation to communication of circadian rhythms between mother and fetus. For instance, pregnant rats exhibit elevated melatonin levels in the night phase compared to non-pregnant females. They return to non-pregnant levels on the second day of parturition. A similar change in profile has been shown for human melatonin patterns [22]. Administration of radio-labeled melatonin to pregnant rats revealed that it is conveyed to the fetus through umbilical circulation [23]. How long melatonin is stable in the fetus requires further investigation. There is, however, strong evidence that maternal melatonin can influence fetal circadian rhythms by binding to melatonin receptors on embryonic day 18 (E18) in rats, one day before neogenesis in the SCN [24], and in humans during the 19th week of gestation [25]. In rats, melatonin receptors are spread throughout fetal tissues and the placenta [26,27]. Studies on melatonin deficient mouse models as well as on pregnant females with lesioned pineal glands [28,29,30], however, show that the embryonal circadian rhythm is still synchronized to the maternal rhythm. This suggests that additional signals are sufficient to synchronize the developing circadian clock in the absence of melatonin.

Glucocorticoids—GCs have been shown to have significant circadian entrainment functions in adult animals [31] and to play an important role in the maturation of numerous fetal organs, in particular of the lung [32,33,34]. High cortisol levels are associated with elevated blood pressure, an increased risk of major depression, and metabolic changes [35,36,37,38]. Similar health effects have been observed in humans and animals when pregnant females were stress challenged. Paradoxically, and similar to melatonin, GC levels gradually increase in pregnant females until the last day of gestation [39]. The maternal levels of GCs in the fetus seem to be precisely controlled by the placenta, as too high and too low GC levels may result in diseases in adulthood [40,41]. The placenta can protect the embryo from excessive GC exposure via 11β-hydroxysteroid dehydrogenase type 2 (11β-HSD2), which catalyzes inactivation of GCs [42]. Expression of the glucocorticoid receptor (GR) in the embryo is highly region specific and changes dynamically along the period of gestation. This dynamic leads to altered local sensitivity of embryonic tissues to GC stimulation [43,44,45,46,47]. In addition to this, certain events can persistently alter GR sensitivity in the developing embryo. Such GR programming may, for example, be caused by elevated stress [48,49,50].

Recent work by Astiz et al. [51] in mice at E17 sheds new light on how GC sensitivity of the fetal hypothalamus can be programmed. The researchers show that hypothalamic diurnal expression rhythms of GR and REV-ERBα, a clock protein that also reduces GR stability [52], are anti-phasic in wild-type and absent in clock deficient mice, implying that the fetal GR response to GCs is controlled by the fetal clock. Interestingly, the presence of GR in the SCN decreases within the first week after birth, just before the animals display robust circadian gene expression in peripheral tissues [53], and the adult SCN no longer responds to GCs [53,54]. This leaves the question as to whether GC levels can influence the stability of the central clock during embryonic development, which may then have implications for the SCN’s response to environmental signals in adulthood.

Cecmanova et al. [55] reported that GCs increase the spontaneous development of rhythmicity in the fetal SCN and entrain the embryonic clock of PER2::LUC circadian reporter mice. The mechanism underlying the effect of GCs on the immature SCN clock is largely unknown. However, it has been suggested that GCs may not use the canonical GR pathway via transcriptional activation through glucocorticoid response elements (GRE), but rather signals via non-genomic cAMP response element-binding protein (CREB)-related signaling pathways that induce c-fos expression [55,56]. Long-term GC induced changes in the circadian clock may involve epigenetic modifications. Maternal treatment with synthetic GCs and prenatal stress during late gestation leave long-lasting changes in gene expression in the offspring, associated with changes in promoter methylation and acetylation at corresponding loci [50,57,58,59]. Intriguingly, these changes are different from those induced by natural GC surges [57,59] implying that synthetic and natural cortisol may differ in their capacity to affect the epigenome.

In adult animals and humans, the adrenal glands receive rhythmic signals from the SCN that, in turn, orchestrate the rhythm of GC secretion [60,61,62]. Similar to adult peripheral oscillators, fetal adrenal clocks receive maternal zeitgeber input through GCs [63]. In capuchin monkeys, the expression of *Per2* and *Bmal1* genes in the adrenal gland resembles respective expression rhythms in the fetal SCN, suggesting that the phase of fetal adrenal gland and SCN are controlled by the same oscillator, namely the maternal SCN. The superiority of the maternal SCN as a synchronizer of the fetal circadian system has been underscored by lesions of the maternal SCN in pregnant rodents. Loss of the SCN leads to desynchronization of behavioral and physiological rhythms in the pups after birth [64]. The successful restoration of wheel running activity in adult rodents with SCN lesions through embryonic transplantation of SCN grafts or cells from mid-gestation provides further evidence of the full functionality of the embryonic SCN clocks [65,66].

Clock gene expression—While the SCN is fully developed during early gestation, clock gene expression in other tissues is at low levels at E15. The fetal pineal gland, for instance, does not express the *Per2* but does express *Bmal1*, *Clock*, and *Cry2* [67]. On the other hand, the fetal liver expresses all clock genes, but only *Rev-erbα* shows oscillation [65]. In contrast to the clock gene expression in fetal tissues, tissue explants of PER2::LUC reporter mice at E15 measured in vitro reveal self-sustained circadian rhythms [68,69]. These rhythms become stronger as the fetus approaches term [67]. Due to these ambiguous results assessed both in vivo and in vitro, it is difficult to conclude whether the detected rhythms reflect the actual circadian clock rhythm in the embryo or if they are a result of the experimental manipulation. Moreover, to what extent these clock genes in various tissues, including the SCN, represent functional circadian clocks is currently unknown. Given that various clock deficient mouse models can be born and live to adulthood with various metabolic and sleep disorders [70,71,72,73,74,75], it can be assumed that the integrity of the circadian system plays an important role in a number of physiological functions, e.g., metabolic balance and sleep homeostasis [64]. The extent to which these physiological mechanisms work in the fetus still has to be shown.

Infancy—While during gestation the embryonal clock is directly synchronized by the maternal SCN, this direct communication is lost after birth. Therefore, right after birth the infant relies on the development of its own independent central clock (or external time signals mediated, e.g., through the mother’s milk—see below). In the first week after birth, the central clock is relatively immature and contains 13% of the adult number of arginine vasopressin (AVP) expressing neurons and only few cells positive for vasointestinal peptide (VIP) [76]. At this developmental stage, the circadian system is substantially vulnerable to environmental and maternal influences. During the first week, the infant brain undergoes extensive changes in neuronal and glial networks into functional circuits [77]. Importantly, core elements of the circadian system, such as the *Per*, *Cry*, *Bmal1* and *Clock* genes, are already expressed in the fetal SCN but do not show a clear circadian rhythm [78,79,80,81]. On day 10 after birth, a clear oscillation of these genes and their proteins is observed [82,83].

Light entrainment—During synaptogenesis, the terminals of the retino-hypothalamic tract (RHT) innervate the ventrolateral region of the suprachiasmatic nucleus (SCN). This process takes place shortly after birth [84]. Intrinsically photosensitive retinal hypothalamic ganglion cells (ipRGCs), which are a key for measuring light irradiance, integration of signals coming from retinal receptors, and their mediation to the SCN, are already detectable at around mid-gestation (E11) in mice [85]. Hence, infants may already receive light information immediately after birth. However, due to their broad distribution in the retina that does not yet contain any classical photoreceptors, image forming vision is not present in mice until postnatal age (P10) [86]. The presence of ipRGCs containing melanopsin, however, enables new-born pups to show negative phototaxis as early as P6 [86,87,88,89], implying that light may also control physiological functions, sleep-wake rhythms, alertness and cognitive functions of infants immediately after birth. In humans, the RHT is detectable at week 36 of gestation [90,91]. This was confirmed in new-born babies kept under constant darkness (DD) or light-dark cycles of 24 h (LD) in the first few days after birth. The plasma of babies kept in DD revealed elevated plasma melatonin levels compared to those kept under LD conditions. This suggests that the neonate is sensitive to light immediately after birth and a connection between RHT, pineal gland and SCN is established. However, whether these light signals induce protein kinase cascades and Per induction in the SCN, as it is the case in adult animals [6,92], is not known to date.

Maternal milk—In addition to light, breast feeding in humans or lactation in rodents is a further important synchronizing parameter of the infant’s circadian system in the first days after birth [93]. Human studies have shown that maternal milk contains several components, which may serve as regulators of the infant’s circadian. Breast milk contains high concentrations of cortisol, tyrosine and immune factors, e.g., cytokines, during the light phase, whereas leptin, melatonin and tryptophan are high during the night phase (Figure 1) [94,95,96,97,98]. Consequently, formula-fed children might show a different development and phase of their circadian clock due to a lack of the maternal humoral components mentioned above. There are only few studies showing that maternal hormones in breast milk may serve as regulators of circadian clock maturation or have an impact on the synchronization of the infant’s clock. Existing studies address the impact of milk supplemented with cortisol or melatonin only. They demonstrate a significant impact of both hormones on the infant’s sleep rhythms, sleep fragmentation and duration [99]. Cubero et al. [100] found that formula-fed children have lower levels of 6-sulfatoxymelatonin in their urine compared to breast-fed infants. This suggests that maternal melatonin secreted into the bloodstream and transferred into maternal milk [97] crosses the infant’s intestinal barrier [101]. Whether melatonin supplementation of formula milk improves sleep in infants via an action on their circadian clock, however, remains unexplored.

Delivery of maternal cortisol to the infant via breast milk has been demonstrated in humans and rodents, showing that not only the diurnal rhythm of maternal cortisol, but also its concentration, is similar in infants [102,103,104,105]. Studies on adult humans and animals reveal that increased GC levels induced by stress or through external administration result in sleep disruption, changed metabolism and stress responsiveness [106,107,108]. Intriguingly, similar effects of cortisol can be observed in infants when maternal milk containing high cortisol levels is consumed [109,110,111,112,113]. This implies that maternal cortisol may play an important role in regulating circadian rhythms and physiological processes in infants. Studies in adult rodents and humans have already demonstrated that GCs affect the clock phase in peripheral oscillators [113]. Moreover, GCs can also directly influence synaptic plasticity [114,115].

## 3. Childhood and Adolescence

Childhood—One of the most prominent physiological outputs of the circadian clock is the sleep/wake rhythm. This has been intensively studied in humans and rodents alike. Sleep is regulated by two processes, the circadian process (process C), and the homeostatic process (sleep pressure or process S). The circadian component of sleep is SCN dependent but also relies on the hormone melatonin, produced in the pineal gland during the dark period. Dim-light melatonin onset (DLMO), which is the start time of melatonin production when decoupled from external light cues, is considered the gold standard for assessing the circadian pacemaker phase in humans [116]. Another tool used by chronobiologists to assess the general clock phase is the Munich chronotype questionnaire [117]. This estimates clock phase by tracking voluntary sleep schedules. Both tools have been used in young children and have been shown to be a reliable measure of circadian rhythms. Young children and toddles differ from adults, as they are predominantly morning chronotypes (i.e., showing earlier bedtimes) [118] with earlier DLMO [119]. They also tend to follow a biphasic sleep pattern, the disappearance of which is one of the milestones of early childhood [120]. Unsurprisingly, regular napping leads to kids falling asleep approx. 1 h later at night, with higher sleep latency and shorter sleep duration when compared to non-napping toddlers [121]. While the children transition out of the biphasic sleep pattern, they may be experiencing sleep loss, as exemplified by an increased slow-wave activity of toddlers deprived of a nap [120]. Children (and teenagers, as we will discuss below) are often forced to function according to their parents’ schedule, or that of their preschool. Misalignment of one’s intrinsic clock and such social schedules—which is usually followed by elongated and shifted sleep schedules during free days, resembling short transcontinental trips on sleep logs—is termed social jetlag (SJL). In fact, kids attending preschool experience higher rates of social jetlag than their home-staying peers (differences of 26.3 vs. 17.6 min in sleep phase between weekdays and weekends) with a quarter of kids’ SJL being greater than 30 min [122]. While larger SJL correlates with health problems [123,124], such a connection has not yet been definitively described for young kids. For example, SJL in children has no significant negative effects on temperament [122]. However, young children entrain to light with high individual sensitivity [125] and, similarly to adults, will phase advance their rhythm when in more natural environments (camping, low light pollution, ref. [126]). This light entrainment may have adverse effects when mistimed, as children sleeping near a screen tend to get around 20 min less sleep [127].

Adolescence—While alterations in sleep patterns of young children may be a cause for concern for their parents, the weekly shifts in sleep patterns seen in many teenagers are significantly worse. As many as 45% of US adolescents may not be getting adequate sleep [128] and as many as 16% may suffer from delayed sleep phase disorder characterized by increased daytime sleepiness and inability to sleep at normal times [129,130]. This is mostly caused by a rapid phase-delay in chronotype which accompanies the onset of puberty and reaches its peak at around 20 years of age [131,132]. While the end of puberty is marked by the cessation of bone growth, the end of adolescence, as proposed by Roenneberg et al. in 2004, could be defined as the point of one’s latest chronotype. Its timing also shows a sex difference, with girls reaching it approx. 1.5 years earlier than boys [132], in line with the earlier onset and completion of puberty in females. Considering that these delays in phase correlate with secondary sex development [131,132,133] and both sleep and the circadian clock are modulated by steroids [134,135,136,137], one could infer a cause–effect relationship. Physiologically, the shift to later chronotypes seems to be caused by a combination of slower sleep pressure build-up and a circadian phase delay [128]. Indeed, a study comparing pre-, early- and post-pubertal children shows that the sleep pressure dissipation rate (measured as the decline in the 2-Hz electroencephalography power band) does not differ, while the build-up of sleep pressure is slower in the older group [138]. This is likely compounded by environmental factors such as limited exposure to light during schooldays coupled with increased exposure to light-emitting electronic devices in the evening [139,140]. On a molecular lever, the physiological phase shift could be driven by chromatin modifications, as it was observed that later sleep timing correlates with methylation levels of circadian genes [141]. While this phenomenon was mostly studied in humans, a few animal studies show that it may similarly apply to other mammalian species. A similar shift of 1 to 4 h in activity rhythms during puberty was observed in macaques, degus, rats, mice and in fat sand rats [128]. Juvenile mice also show differences in their entrainment capacities [142].

Late chronotypes, which teenagers predominantly are, tend to sleep less on average, experience greater SJL and compensate for lost sleep on weekends [143]. This may affect their physical and mental health. As a consequence, researchers have recommended that school start times should be delayed to fit the natural rhythm of adolescents [144,145]. A pilot study which shifted the school start time from 8:50 AM to 10:00 AM showed a 12% improvement in academic progress and 50% drop in absences due to illness [146]. Additionally, to mitigate the harmful effects of blue light exposure, a week of blue light blocking glasses was shown to attenuate LED-induced melatonin suppression and subjective alertness before bedtime [147].

## 4. Menopause

Reproduction in mammals only lasts for a limited time [148]. This applies to females and, to a smaller extent, to males. Both sexes, however, experience physiological changes during the aging process related to changes in sex hormone levels [149]. The onset of menopause is often described as the end of a woman’s “biological clock” [150,151]. When the ovaries start running out of egg follicles that release estrogen, they also become less responsive to other hormones that stimulate ovulation. As a result, the ability of females in terms of reproduction rapidly drops to zero, whereas andropause in men is characterized by a rather gradual reduction in testosterone levels over decades [152]. This testosterone level reduction, however, has only little effect on the viability of sperm cells and, thus, the principal ability to reproduce [153,154,155].

Clock of the reproductive system—The discovery of the ovarian clock [156,157] considerably changed the perspective on reproduction in women. The ovarian clock is controlled by neuroendocrine signals from the SCN [158]. A substantial body of evidence from human and mouse studies shows that the circadian clock plays a crucial role in the physiological processes of the reproductive system, such as ovulation and hormone secretion. Deficiencies in the circadian clock through clock gene mutations can lead to impaired reproductive success [159]. For example, female mice with Bmal1 deficiency or double clock mutations in *Per* and *Cry* exhibit disrupted mating behavior and infertility [160,161,162,163,164,165,166,167]. Single clock gene mutations in *Per1* and *Per2* lead to reduced ovarian function [160,168,169]. Conversely, other studies in *Bmal1* knockout and *Clock* mutant mice show that a deficiency in clock genes weakens the luteinizing hormone (LH) surge, but ovulation is unaffected [71,164,170,171]. Thus, the circadian rhythm appears to play a crucial role in determining the timing of the LH surge, but it is not necessary for spontaneous ovulation. The local clock in the ovaries, however, has a significant impact on the timing of the ovulatory response to LH [172]. This suggests that the ovarian clock may control LH receptor signaling and ultimately influence the timing of ovulation.

HPG axis—In female rodents during the ovulatory cycle, sex steroid secretion is controlled similarly to that in males via the hypothalamus-pituitary-gonad (HPG) axis (Figure 2). It initiates from a neuroendocrine cascade, e.g., gonad releasing hormone (GnRH) in the medial preoptic area (mPOA), kisspeptin neurons in the anterior ventral paraventricular nucleus (AVPV), and arginine vasopressin (AVP) neurons in the SCN. This dictates the release of luteinizing hormone (LH) and follicle stimulating hormone (FSH) in the pituitary gland [158,173] (Figure 2). Both hormones act on gonads and induce gametogenesis and sex hormone production. During the follicular phase of the menstrual cycle, FSH stimulates the maturation of ovarian follicles and the secretion of estradiol. When estrogen levels consistently peak for 48 h in a human, the secretion of FSH is suppressed, leading to a surge in GnRH from the hypothalamus. This GnRH surge stimulates the release of gonadotropic hormones, including a surge in LH. The combination of the FSH peak and LH surge triggers ovulation. Following ovulation, FSH levels remain low, preventing the growth of additional follicles [174]. Additionally, they modulate the clock gene rhythm in the ovarian tissue [175,176].

Aging female SCN—The activity of the HPG axis is regulated by three crucial brain nuclei, i.e., the mPOA, the AVPV, and the SCN, which control reproductive success. The activity of these nuclei is also regulated by the balance of positive and negative effects of sex steroids [158]. The mechanism mediating both negative and positive feedback of estradiol is complex and still not fully understood. The abnormally high or persistently low estrogen plasma levels during reproductive senescence can impact HPG axis activity, resulting in reduced ovulation [177]. Interestingly, the changes in GnRH and kisspeptin secretion in HPG axis vary among species. In humans, it has been shown that high plasma concentrations of estrogen are associated with insensitivity of the HPG axis to estrogen feedback [178,179]. This, in turn, leads to an increase in GnRH [180,181]. It is important to note that these changes in HPG axis activity are caused by depletion of ovarian follicles in middle-aged women. In contrast to humans, aging acyclic rodents retain the follicles in their ovaries [182,183,184]. Instead, they show a reduction in hypothalamic GnRH cell numbers followed by alterations in LH surge that contribute to reproductive senescence [185,186,187]. The importance of the hypothalamus in regulating reproductive ability could be confirmed with transplantation experiments. Aged ovaries transplanted into young adult ovariectomized female rats, for example, result in restoration of ovulation [188,189]. Besides these differences, in both humans and rodents, the rise of FSH concentrations is a main feature of reproductive senescence [190,191]. Altered LH secretion patterns characterized by an increased duration and decreased frequency of LH pulses can also be observed in both acyclic rodents and premenopausal women [192,193]. Therefore, to fully understand the transition to menopause in humans, the use of rodent models in hypothalamic decline can be of great help.

**Figure 2 biology-12-00383-f002:**
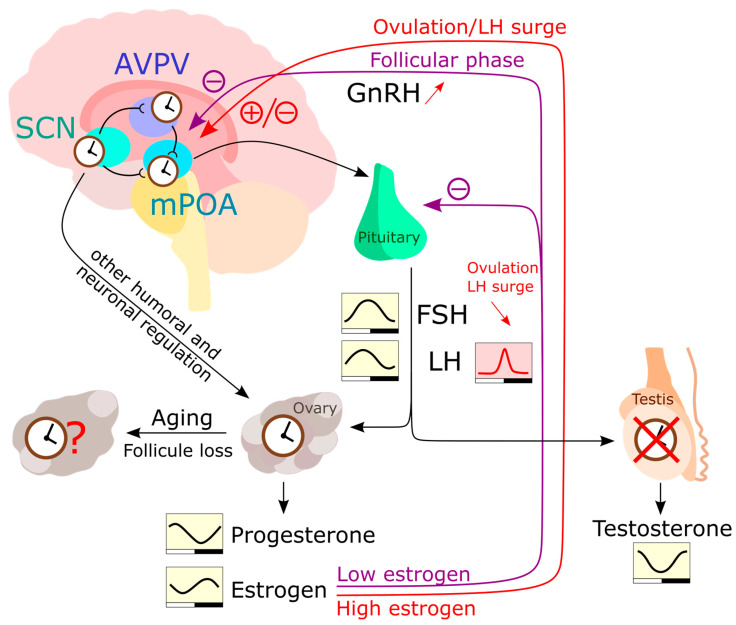
The HPG (hypothalamic-pituitary-gonadal) axis regulates the levels of sex hormones in the human body. In the hypothalamus, the AVPV and mPOA communicate with each other and receive input from SCN neurons. After stimulation with peak estrogen, mPOA neurons produce GnRH which is transported to the pituitary. The pituitary then produces LH and FSH in response to GnRH stimulation, which act on the gonads to induce the production of sex hormones. Unlike the testes, the ovaries have been reported to express a functional circadian clock and respond to humoral and neuronal signals independently of LH and FSH rhythms. The rhythmic production of estrogen and progesterone is therefore likely influenced by the peripheral ovarian clock, as well as by rhythmic LH and FSH levels [194]. Estrogen also can inhibit GnRH secretion and gene expression. Testosterone is produced in a rhythmic manner [195], but this rhythmicity is largely HPG axis driven. During a woman’s life, the number of follicles in the ovaries diminishes until extremely low follicle numbers trigger menopause. However, the influence of this process on the peripheral ovarian clock is yet to be fully understood.

A very noticeable alteration associated with aging is changes in sleep–wake rhythms that are mainly, but not exclusively, regulated by the SCN [196,197,198]. A large body of evidence shows that age-related changes in the SCN comprise reduced neuronal activity of AVP, and VIP neurons important for intercellular coupling within. and robustness of, the circadian pacemaker structure. Loss of AVP/VIP activity contributes to the decline in behavioral and physiological rhythms [199,200,201]. Both neurons have also been linked with control of ovulation [173,202,203,204]. For instance, AVP, that mediates SCN output, triggers the LH surge through activating kisspeptin neurons in AVPV, which project to the mPOA [205,206]. VIP neurons, in turn, project to the mPOA and control the timing of the LH surge via the triggering of GnRH neurons [206]. According to Krajnak et al. [186] and Davidson et al. [207], the molecular clock rhythm itself does not change significantly in aged animals. Therefore, it can be postulated that GnRH and kisspeptin circuits may become less sensitive during the aging process. Further research into the role of the circadian clock system in neural circuits regulating LH and FSH secretion and ovulation is important for understanding timing mechanisms contributing to reproductive senescence.

## 5. Old Age

With increasing age, many physiological processes lose integrity, which results in a loss of resilience when faced with environmental challenges [208]. This increases the vulnerability to disease and, ultimately, death. One’s longevity is only partially (~15–30%) [209] explained by genetics, with a marked influence of personal history and life-long habits [210]. Between species, longevity varies largely. Across 26 different mammalian species with diverse lifespans, thousands of genes correlate with longevity. These can be divided into negatively correlated (mainly involved in energy metabolism and inflammation) and positively correlated (associated with DNA repair, microtubule organization and RNA transport). Remarkably, many genes in the negatively correlated group are under tight circadian regulation, suggesting that one adaptive value of the clock may lie in avoiding persistently high expression of such death promoting genes [211]. In addition to this, many ailments typically associated with old age also have a circadian component. These include neurodegeneration [212], cancer [213], cardiovascular diseases [214], hypertension [215] and others. Other dysfunctions include poor sleep [216] and poor cognition [217].

Aging both reduces the amplitude and phase advances of the circadian rhythms of the body, including sleep, body temperature, cortisol and melatonin [218,219,220,221] (Figure 3). These changes can be linked to circadian rhythm perturbation in both the SCN and peripheral tissues. Transplant experiments demonstrate the importance of the SCN in this context, as old animals that receive transplants of fetal SCN improve their rhythms of locomotor activity, body temperature, water consumption, pro-opiomelanocortin, corticotropin releasing hormone, and even show increased longevity [222,223,224]. More recent studies show how expression of neurotransmitters in the SCN diminishes with age. Both AVP and VIP levels are reduced in aged humans [225,226,227] and rodents [199,228]. In addition, GABAergic synapses also diminish in numbers in aged mice [229]. This understandably leads to lower amplitudes and lower levels of spontaneous firing activity [230,231,232]. Overall, this suggest that an aging SCN loses its internal synchrony [233], resulting in a loss of overall coherence of SCN outputs [234].

The disruption of SCN rhythms may partially be driven by weakened entrainment, with a focus on light. This can be caused by physiology and environmental factors alike. However, older people tend to spend more time exposed to light than younger adults [235]. That may be offset by degenerative changes in the eye, which loses the transmissibility of its lens and pupil area with age [236,237] and therefore communicates lower levels of light via the RHT to the SCN. This is estimated to amount to a 72% loss of transmissibility at 480 nm (the absorption maximum for circadian entrainment) from the age of 10 to 80 years [236] (Figure 3). In addition, the RHT communicates with the SCN via α-amino-3-hydroxy-5-methyl-4-isoxazolepropionic acid (AMPA) and N-methyl-D-aspartate receptors (NMDA). The latter have been reported to show response deficits in older mice [237].

Downstream of the SCN, peripheral clocks are also disturbed in the elderly (Figure 3). Transcriptional rhythms of multiple organs have been reported as attenuated in aged patients and mice, including ovaries [238], thyroid [239], skeletal and vascular muscle [240] and kidneys [241]. Some of those peripheral clock rhythms could potentially be restored by timed exercise [240,241]. Studies in mice show that voluntary wheel running can restore dysfunctions in activity rhythms in older mice, and there seems to be a sex difference in this response, as older females run significantly longer distances than males [242]. Surprisingly, some of these studies show a relative retention of liver rhythms [238,239], while another study conducted in rats suggest that changes in liver circadian genes during aging might be characteristic of males [243].

Mouse wheel-running studies also report that older animals have a lower activity with a delayed onset (i.e., a phase delay) [242]. This is accompanied by an increase in onset variability [244]. Interestingly, this disturbance of wheel-running patterns seems to precede learning and memory impairment, suggesting that circadian rhythm disturbances could serve as an early predictor of cognitive decline [245]. Similar processes may be at play in humans as, in an accelerometry study of thousands of US adults, more advanced biological aging was linked to lower amplitude, less daily stability and higher inter-day variability of rest–activity rhythms [246].

Elderly subjects also experience significant changes in their chronotype and sleep patterns. In line with a phase advance of physiological rhythms, chronotype shifts towards earlier times with advancing age until it surpasses the average (early) chronotype of children [117]. This may be driven by endocrine changes, as is suspected of the late-chronotype shift observed in teenagers. Worryingly, around 50% of old people suffer from poor sleep [247]. This is associated with changes in sleep architecture which include longer latencies to fall asleep and less time spent in deep non-REM sleep (stages 3 and 4), as well as REM sleep [248,249]. In addition to an effect on well-being, self-reported short sleep duration correlates with poor cognitive performance [250]. To mitigate this, several pilot studies propose a use of bright light in nursing homes to improve sleep (among other symptoms) in elderly patients, and the results are promising [251,252,253].

It is not yet elucidated whether sleep disturbances are a consequence of aging or an aggravating factor. Nevertheless, research suggests that, in older patients, different forms of chronotherapy could improve well-being and health. Further studies are needed to assess the effectiveness of such interventions.

## 6. Conclusions

In conclusion, we observe dramatic changes in circadian architecture along the life cycle in humans, as well as in experimental rodents. While circadian rhythms are initiated under control by maternal factors inside the womb, they only fully mature after birth. During older age, rhythms gradually become more and more disrupted due to a lack of internal synchronization at all levels of organization—from genes to behavior—and, at the same time, a loss in responses to external zeitgeber input. These changes, while being interesting in themselves from a biological (evolutionary) perspective, may also be of high value for medical reasons. Circadian rhythms affect disease resistance and development. They impinge on diagnosis parameters and may affect the effectiveness of treatments. Thus, considering the age of a patient may critically affect the best time (and, potentially, choice) of therapy. One special case is fertility and reproduction. Clocks have been shown to play important roles in both male and female reproductivity, and alterations in circadian rhythms may indicate imminent changes in reproductive fitness. On the other hand, circadian phenotyping may improve interventions aiming at increasing reproductive success, especially in relation to increased age. Finally, it will be of interest to study circadian rhythms along the life cycle of extremely short or long-lived species to better understand and potentially exploit the interaction between circadian and lifetime clocks.

## Figures and Tables

**Figure 1 biology-12-00383-f001:**
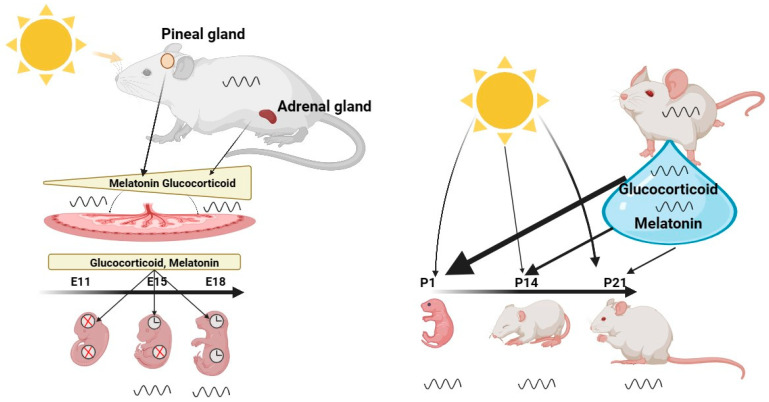
The Role of the Maternal Clock in Synchronizing the Clock of the Embryo and Infant. Maternal signals, such as melatonin and glucocorticoids, gradually increase during the gestation period and synchronize the circadian clock of the embryo. These hormones are transmitted through and controlled by the placenta, which serves as the interface between the fetal and maternal circulatory systems. Their function is to adapt the developing embryo to the external environment, ensuring optimal preparation for life outside the womb. Central and peripheral clocks exhibit different developmental states, with the central clock exhibiting rhythmicity as early as E13, while the peripheral clocks only become rhythmic at developmental age of E18.After birth, the infant is no longer directly synchronized by the maternal suprachiasmatic nucleus (SCN), but relies largely on its own independent clock, which can be directly synchronized by light. However, breast milk, illustrated by a blue drop, contains hormones such as melatonin and glucocorticoids that can modulate the synchronization of the infant’s clock, particularly in the first few days after birth. Once the infant’s retina is fully developed at P13, light becomes a stronger synchronizer than maternal milk. Created with BioRender.com.

**Figure 3 biology-12-00383-f003:**
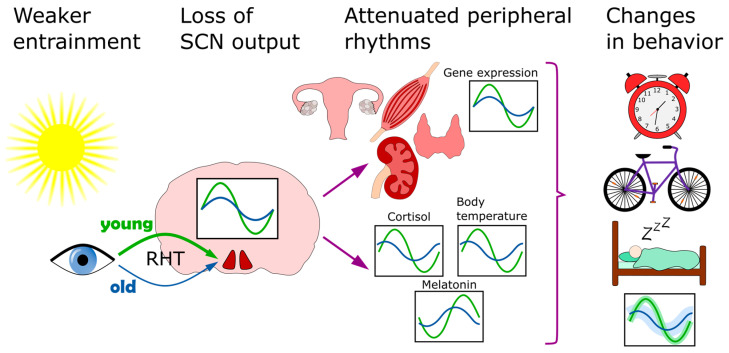
Circadian clock of the elderly. This affected on multiple levels. The entrainment of the clock by light becomes less efficient because of diminished light transmissibility of the lens. In the SCN, the neurotransmitter levels go down and overall coherence is disturbed. Downstream, the lower clock amplitude is also observed in several peripheral organs, including the skeletal muscle, kidney, thyroid and ovaries. As a result of these, and other, changes, the behavioral patterns of sleep/wake and activity also shift.

## Data Availability

Not applicable.

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
