# Peer review of "Circle(s) of Life: The Circadian Clock from Birth to Death"

_biology, 2023, doi:10.3390/biology12030383_

Round 1
Reviewer 1 Report
In this article, the authors provide a comprehensive review of the physiology of the circadian clock across different mammalian stages of development from birth to old ages. The information is well-summarized and relevant to a broad audience. A few recommendations are listed below to easy the reading and provide quick effective information to the reader.
a- As currently shown, Figure 1 is not very informative. The text is very reach in information that is not represented in Figure 1. For example, Fig.1 would be more useful if summarizes the content presented in lines 80 to 127 in some detail. In addition, a schematic representation of embryo development showing key developmental stages E15, E19 etc would aid the reader to match the information in the text to the developmental stage.
b- This reviewer is confused about two statements. In lines 53-55 the authors state “…. studies on female mice with double clock gene mutations showed that the fetal clock rhythm remains unchanged (15). This indicates that the pups are capable of developing autonomous circadian rhythms even without central and maternal clocks.” However, in lines 130-133, the authors state “ In capuchin monkeys, the expression of Per2 and Bmal genes in the adrenal gland resembles respective expression rhythms in the fetal SCN, suggesting that the fetal adrenal gland and SCN are controlled by the same oscillator, namely the maternal SCN.” Some clarification is needed.
c- There is substantial information in the first section and, maybe, there is a need to divide the no role of the maternal SCN in the development of the pup circadian rhythms vs the later need of the SCN for synchronization. May be using subheadings within the “prenatal and infancy” section to differentiate both roles.
d- Fig.2: As it was the case with Fig. 1, Fig.2 does not faithfully describe the content of the text and oversimplifies information to the point that can be misinterpreted. First, the cartoons that represent the different organs/glands need to be labeled. Second, the information from lines 318 to 326 is not represented in the figure. For example, the feedback loops that regulate the release of LH and the surge of GnRH are not represented. As presented in Fig. 2, one can believe that LH leads to E2 and it is not clear what happens after, how the system goes back to normal, the relevance of the feedback loops, the suppression of FSH etc. In addition, the text does not mention progesterone and its role in the process. Maybe the author should separate what it is going on in the follicular phase from what it happens in the ovulation phase.
e- Line 426: Delete “me” in “aging me be characteristic to males”. Replace by “might”? (before ref. 241)
Author Response
Response to Reviewer 1 comments
We would like to thank the reviewer for the positive assessment of our review and are happy to implement their useful suggestions.
Specific points are addressed below:
In this article, the authors provide a comprehensive review of the physiology of the circadian clock across different mammalian stages of development from birth to old ages. The information is well-summarized and relevant to a broad audience. A few recommendations are listed below to easy the reading and provide quick effective information to the reader.
- As currently shown, Figure 1 is not very informative. The text is very reach in information that is not represented in Figure 1. For example, Fig.1 would be more useful if summarizes the content presented in lines 80 to 127 in some detail. In addition, a schematic representation of embryo development showing key developmental stages E15, E19 etc would aid the reader to match the information in the text to the developmental stage.
Reply: The figure was redrawn and now includes information about the developmental stages of a mouse.
- This reviewer is confused about two statements. In lines 53-55 the authors state “…. studies on female mice with double clock gene mutations showed that the fetal clock rhythm remains unchanged (15). This indicates that the pups are capable of developing autonomous circadian rhythms even without central and maternal clocks.” However, in lines 130-133, the authors state “ In capuchin monkeys, the expression of Per2 and Bmal genes in the adrenal gland resembles respective expression rhythms in the fetal SCN, suggesting that the fetal adrenal gland and SCN are controlled by the same oscillator, namely the maternal SCN.” Some clarification is needed.
Reply: We would like to thank the reviewer for pointing out that this phrasing could be confusing for the reader. To clarify: while the maternal clock is not needed for offspring clock development, it does influence its phase. This distinction was emphasized by adding “phase of” in the sentence: “In capuchin monkeys, the expression of Per2 and Bmal1 genes in the adrenal gland resembles respective expression rhythms in the fetal SCN, suggesting that the phase of fetal adrenal gland and SCN are controlled by the same oscillator, namely the maternal SCN.” (Line 152-155)
- There is substantial information in the first section and, maybe, there is a need to divide the no role of the maternal SCN in the development of the pup circadian rhythms vs the later need of the SCN for synchronization. May be using subheadings within the “prenatal and infancy” section to differentiate both roles.
Reply: We thank the reviewer for this comment. Subheadings were now added in the “Prenatal and Infancy” as well as other sections.
- 2: As it was the case with Fig. 1, Fig.2 does not faithfully describe the content of the text and oversimplifies information to the point that can be misinterpreted. First, the cartoons that represent the different organs/glands need to be labeled. Second, the information from lines 318 to 326 is not represented in the figure. For example, the feedback loops that regulate the release of LH and the surge of GnRH are not represented. As presented in Fig. 2, one can believe that LH leads to E2 and it is not clear what happens after, how the system goes back to normal, the relevance of the feedback loops, the suppression of FSH etc. In addition, the text does not mention progesterone and its role in the process. Maybe the author should separate what it is going on in the follicular phase from what it happens in the ovulation phase.
Reply: The figure was updated to represent cycle-specific feedback. Organs and glands are labeled.
- Line 426: Delete “me” in “aging me be characteristic to males”. Replace by “might”? (before ref. 241)
Reply: We apologize for the typo. The change was implemented as suggested. (Line 450)
Reviewer 2 Report
The review of Olejniczak et al. nicely present the age-related differences of the circadian clock. The manuscript is well written and the figures display the main points.
Therefore, I have only minor suggestions.
Page 1, Line 39: “feed the clock” sounds sloppy. Please use another term for example: influences or synchronize……
Page 4, Line 165: The maturation of clock gene proteins was already shown by Ansari et al., in 2009. You might also cite this paper: Ansari, N., et al. (2009). Differential maturation of circadian rhythms in clock gene proteins in the suprachiasmatic nucleus and the pars tuberalis during mouse ontogeny. European Journal of Neuroscience, 29(3), 477-489.
Page 6, Line 259: There are some better citations for the statement that sex development, sleep and the circadian clock probably correlate that 124. Example: Hagenauer and Lee (2012). The neuroendocrine control of the circadian system: adolescent chronotype. Frontiers in neuroendocrinology, 33(3), 211-229.
Author Response
Response to Reviewer 2 comments
We would like to thank the reviewer for this positive assessment of our review and are happy to implement their useful suggestions.
Specific points are addressed below:
The review of Olejniczak et al. nicely present the age-related differences of the circadian clock. The manuscript is well written and the figures display the main points.
Therefore, I have only minor suggestions.
Page 1, Line 39: “feed the clock” sounds sloppy. Please use another term for example: influences or synchronize……
Reply: Line 53/54 was rephrased to now read: “synchronize the clock”
Page 4, Line 165: The maturation of clock gene proteins was already shown by Ansari et al., in 2009. You might also cite this paper: Ansari, N., et al. (2009). Differential maturation of circadian rhythms in clock gene proteins in the suprachiasmatic nucleus and the pars tuberalis during mouse ontogeny. European Journal of Neuroscience, 29(3), 477-489.
Page 6, Line 259: There are some better citations for the statement that sex development, sleep and the circadian clock probably correlate that 124. Example: Hagenauer and Lee (2012). The neuroendocrine control of the circadian system: adolescent chronotype. Frontiers in neuroendocrinology, 33(3), 211-229.
Reply: We thank the reviewer for suggesting these pertinent references. Both Ansari et al., 2009 (ref. [81] in line 189) and Hagenauer and Lee, 2012 (ref. [137] in line 284 ) are now cited.
Reviewer 3 Report
This review manuscript by Olejniczak et al. is a quite detailed and interesting description of both historical and state of the art data about lifelong changes in the circadian system and in its synchronisation mechanisms, from fetal life to old age. The review covers a wide range of knowledge in the field (although synchronisation by light is less thoroughly described).
The review includes regulation of reproduction throughout life and also extensive data about glucocorticoids which is at the expense of the clarity of the account. I would suggest to use more subheadings to better organize the whole story and make it easier to read.
Another point concerns the illustrations: the review would be improved with a same style for the design of figures (notably Fig1 vs Fig2 and 3). For instance illustration of rhythmic functions (sine waves) should have the same design (also the light/dark cycle) and the brain shapes (fig 1 and 2) should be similar.
Author Response
Response to Reviewer 3 comments
We would like to thank the reviewer for this positive feedback and helpful suggestions. Below, we addressed the concerns point-by-point:
This review manuscript by Olejniczak et al. is a quite detailed and interesting description of both historical and state of the art data about lifelong changes in the circadian system and in its synchronisation mechanisms, from fetal life to old age. The review covers a wide range of knowledge in the field (although synchronisation by light is less thoroughly described).
Reply: While the process of synchronization by light is an important aspect of clock function across lifetime, a detailed description of light entrainment is a vast topic that merits a review of its own. Indeed, two excellent reviews on this subject were recently published (von Gall, 2022, doi: 10.3390/ijms23052778; Ashton, Foster and Jagannath, 2022, doi: 10.3390/ijms23020729). To anticipate the need of a curious reader, we now cited these two reviews in line 53.
The review includes regulation of reproduction throughout life and also extensive data about glucocorticoids which is at the expense of the clarity of the account. I would suggest to use more subheadings to better organize the whole story and make it easier to read.
Reply: We are grateful for this comment. To improve upon readability of the paper, we’ve now added subheadings in the “menopause” as well as other sections.
Another point concerns the illustrations: the review would be improved with a same style for the design of figures (notably Fig1 vs Fig2 and 3). For instance illustration of rhythmic functions (sine waves) should have the same design (also the light/dark cycle) and the brain shapes (fig 1 and 2) should be similar.
Reply: Figure 1’s style was improved upon, and additional information were added concerning the embryonal and postnatal developmental stages.